

# Data-flow-based adaption of the System-Theoretic Process Analysis for Security (STPA-Sec)

Jinghua Yu[1,2], Stefan Wagner[2] and Feng Luo[1]

[1] School of Automotive Studies, Tongji University, Shanghai, China
[2] Institute of Software Engineering, University of Stuttgart, Stuttgart, Germany

## ABSTRACT

Security analysis is an essential activity in security engineering to identify potential system vulnerabilities and specify security requirements in the early design phases. Due to the increasing complexity of modern systems, traditional approaches lack the power to identify insecure incidents caused by complex interactions among physical systems, human and social entities. By contrast, the System-Theoretic Process Analysis for Security (STPA-Sec) approach views losses as resulting from interactions, focuses on controlling system vulnerabilities instead of external threats, and is applicable for complex socio-technical systems. However, the STPA-Sec pays less attention to the non-safety but information-security issues (e.g., data confidentiality) and lacks efficient guidance for identifying information security concepts. In this article, we propose a data-flow-based adaption of the STPA-Sec (named STPA-DFSec) to overcome the mentioned limitations and elicit security constraints systematically. We use the STPA-DFSec and STPA-Sec to analyze a vehicle digital key system and investigate the relationship and differences between both approaches, their applicability, and highlights. To conclude, the proposed approach can identify information-related problems more directly from the data processing aspect. As an adaption of the STPA-Sec, it can be used with other STPA-based approaches to co-design systems in multi-disciplines under the unified STPA framework.

## INTRODUCTION

System security is an emergent property of the system, which represents a state or condition that is free from asset loss and the resulting loss consequences. System security engineering, as a special discipline of system engineering, coordinates and directs various engineering specialties to provide a fully integrated, system-level perspective of system security and helps to ensure the application of appropriate security principles and methodologies during the system life cycle for asset protection (*Ross, McEvilley & Oren, 2016*). Violating system security constraints causes unexpected incidents, like mission failure or leaking sensitive information, and finally leads to financial or even life losses. Therefore, security needs to be considered carefully in system design. Security requirement analysis, referring to the activity of analyzing systems in security-related aspects to

Corresponding author
Jinghua Yu, yujinghua@tongji.edu.cn

achieve security requirements in this research, is performed in the early security engineering phase and helps to manage system risks and make decisions.

Traditional security analysis approaches, being designed for former relatively simple systems, are not effective to analyze increasingly complex systems. For example, a modern vehicle is a Cyber-Physical System, which consists of not only tens of thousands of physical components but also large amounts of software codes. A vehicle Over-The-Air software update system, as a Socio-Technical System, refers to not only the technical parts but also social entities like data providers and regulations. However, most traditional approaches start with system decomposition and analyze the components independently, which leads to overlooking the impacts of interactions among components. Besides, traditional causality models attribute accidents to an initial component failure cascading through a set of other components (like dominos) (*Young & Leveson, 2014*) and cannot address causes of losses with non-linear cause-and-effect linkages.

To meet the requirements of modern systems, a relatively new approach for safety engineering called System-Theoretic Process Analysis (STPA) was proposed (*Leveson & Thomas, 2018*) and its extension for security named STPA-Sec was presented later (*Young & Leveson, 2013*). However, the STPA-Sec does not consider non-safety but security-critical issues (e.g., data confidentiality) and lacks efficient guidance for identifying information security concepts.

Therefore, we propose a data-flow-based adaption of the STPA-Sec (named STPA-DFSec) to overcome the mentioned STPA-Sec's limitations. The analysis process of a vehicle digital key system is presented to demonstrate how to use the proposed approach. We also analyze the same system by using the original STPA-Sec and compare their outcomes. Finally, we discover the relationship between concepts in both approaches and conclude the highlights and applicability of them.

The rest of this article is organized as follows. In the "Related Work" section, we introduce the established approaches and the STPA series with research gaps. In the "Methodology" section, we briefly describe the STPA and STPA-Sec approaches and propose the adaption in detail. In the "Case Study" section, we present the analysis processes of an example case by using both original and adapted approaches to demonstrate how to use them and make the comparison. Finally, we summary this article in the "Conclusion" section.

## RELATED WORK

### Established security requirement analysis approaches

We compare established approaches (other than the STPA-based ones) for security requirement analysis from several industry guidelines, like SAE J3061 (*SAE, 2016*) in the automotive industry and NIST cybersecurity framework for critical infrastructure (*NIST, 2018*), as well as other published research. Note that not all approaches use the term "requirements" explicitly. We regard all the activities that aim to identify security requirements as relevant approaches. For example, threat analysis and risk assessment (TARA) is a typical activity for analyzing security problems. The outputs of TARA are the

**Table 1 Summary of established security analysis approaches other than STPA-based ones.** Brief Introduction of established security analysis approaches (other than STPA-based ones) with their categories.

| Approach | Brief introduction | Category |
|---|---|---|
| NIST cybersecurity framework method (NIST, 2018) | Cybersecurity Framework is a risk-based approach to managing cybersecurity risks of critical infrastructure published by the National Institute of Standards and Technology (NIST). Five functions of the framework core are "Identify", "Protect", "Detect", "Respond", and "Recover" | Threat-oriented; Component-based |
| EVITA TARA process (Ruddle et al., 2009) | EVITA TARA method was proposed in the E-Safety Vehicle Intrusion Protected Applications (EVITA) project, which aims to design, verify, and prototype a secure architecture for automotive on-board networks | Threat-oriented; Scenario-based |
| TVRA process (ETSI, 2017) | Threat, Vulnerabilities, and implementation Risks Analysis (TVRA) is a process-driven TARA methodology developed by the European Telecommunications Standards Institute (ETSI) | Threat-oriented; Component-based |
| OCTAVE Allegro (Caralli et al., 2007) | OCTAVE Allegro is a streamlined approach for information assets, as an agile variant of the Operationally Critical Threat, Asset, and Vulnerability Evaluation (OCTAVE), which was developed by the Software Engineering Institute and sponsored by the U.S. Department of Defense | Threat-oriented; Component-based |
| HEAVENS TARA process (Olsson, 2016) | HEAling Vulnerabilities to ENhance Software Security and Safety (HEAVENS) is a systematic approach of deriving security requirements for vehicle E/E systems, including processes and tools supporting for TARA | Threat-oriented; Scenario-based |
| FMVEA (Schmittner et al., 2014) | Failure Mode, Vulnerabilities and Effects Analysis (FMVEA) is an approach evolved from the Failure Mode and Effect Analysis (FMEA) to identify vulnerability cause-effect chains, which consists of vulnerabilities, threat agent, threat mode, threat effect, and attack probability | Threat-oriented; Component-based |
| CHASSIS (Raspotnig, Karpati & Katta, 2012) | Combined Harm Assessment of Safety and Security for Information Systems (CHASSIS) is a unified process for identifying hazardous scenarios by using UML-based models (misuse cases and sequence diagrams) | System-oriented; Scenario-based |

source of security requirements. Table 1 summarizes the investigated approaches with brief introductions and categories.

With regard to the starting point, most of the mentioned approaches are threat-oriented. They start with identifying threat-related items (e.g., threat source or attack interface) of the system assets or operation scenarios. This is from the point of view of an attacker and aims to protect the system by analyzing and handling all enumerated external threats. Another type is the system-oriented one, which starts with analyzing the system features (incl. structure, function or use case) and tries to find vulnerabilities of the system. Threat-oriented approaches are well-structured and have been widely used in various industries. It is efficient to protect the system against known threats based on a threat database and expert experience. However, threat-oriented ways are not efficient for relatively new systems with less previous experience, and may overlook new kinds of threats. By contrast, the system-oriented approaches are more likely to handle such situations by identifying system vulnerabilities and focusing on controlling potential vulnerabilities relying less on the threat database. Besides, since the external threats are continuously developing, the system-oriented ways are more efficient to ensure the system operation without being compromised regardless of the source and type of threats, just like defending a castle by reinforcing walls and not caring who is the enemy. Furthermore, the system-oriented approaches are more useful for high-level decisions since they view the issues from the perspective of the whole system.

With regard to the basic analysis object, the mentioned approaches can also be divided into component-based and scenario-based classes. The former class views the system as the composition of a set of assets and aims to protect them to achieve the system security, while the latter class focuses on the functional operation of a system and aims to ensure the expected system behaviors. The component-based approaches can protect essential components well and are convenient for the development management since different teams can be responsible for certain components. However, such approaches lack the consideration of the interaction among components. Each component can be secure but attacks may still happen during the interactions. By contrast, the scenario-based approaches consider the interaction among components and focus on providing secure services instead of protecting system components. Such approaches require a global design consideration and more management efforts for cooperation between different development teams.

The previously mentioned approaches are at the process or framework level. Many concrete techniques are used in practice when applying these frameworks. For example, HEAVENS and FMVEA use Microsoft's STRIDE model (*Microsoft, 2009*) to identify potential threats. The attack tree analysis is used in the EVITA process to analyze attacks in depth and obtain attack scenarios (*Ruddle et al., 2009*). The Threat and Operability Analysis (THROP) can be used in the threat identification phase when applying the EVITA process at the feature level (*SAE, 2016*). Since the proposed approach in this article is at the framework level, techniques used in certain steps are only listed here as examples without further investigation.

## System-theoretic process analysis based approaches and highlights

STPA is a hazard analysis approach based on the System-Theoretic Accident Model and Process (STAMP), which views losses as results from interactions among various system roles that lead to violations of safety constraints and analyzes issues at the strategy level. STPA provides a powerful way to deal with complexity by using traceable hierarchical abstraction and refinement (*Young & Leveson, 2014*).

Other than safety engineering, STPA has also been extended into other fields with the same system-theoretic thought. *Young & Leveson (2013)* presented STPA for Security (STPA-Sec), which shares similar steps with STPA and focuses on controlling system vulnerabilities instead of avoiding threats. To perform co-analysis of safety and security under the STPA framework better, *Friedberg et al. (2017)* proposed a novel analysis methodology called STPA-SafeSec, which integrates STPA and STPA-Sec into one concise framework and overcomes limitations of original approaches (e.g., no considerations about non-safety security issues) by introducing security constraints and mapping abstract control structures to real components. *Shapiro (2016)* proposed STPA for Privacy (STPA-Priv), which extends STPA into privacy engineering by introducing privacy concepts and considerations into the general STPA process steps.

The most significant highlight of STPA-based approaches is that they shift from focusing on preventing failures and avoiding threats to enforcing safety constraints and controlling system vulnerabilities. Controlling system vulnerabilities rather than reacting

to threats is more efficient to ensure system security because controlling a vulnerability may reduce the attack risk of several threats. Another highlight is that the STPA-based approaches focus more on the strategy level rather than the tactic level. The tactics are means to accomplish a specific action and are focused on physical threats, while the strategy is regarded as the art of gaining continuing advantages and is focused on abstract outcomes (*Young & Leveson, 2014*). The strategy view is beneficial to broaden the mind and take more aspects like organizational and managerial ones into account. Therefore, the STPA-based approaches are applicable for socio-technical systems and suitable for today's complex systems. Not only physical system components but also humans, natural or social environment and their interactions are all within the scope of the STPA-based approaches. Furthermore, due to the numbers of extensions of STPA in various disciplines, it is easier to perform co-design under the same STPA framework.

## STPA-Sec applications and gaps

The STPA-based security analysis approach (STPA-Sec) has been used to identify system security constraints in various industries. *Khan, Madnick & Moulton (2018)* demonstrated the implementation of STPA-Sec to identify security vulnerabilities of a central utilities plant gas turbine use case in industrial control systems. *Mailloux et al. (2019)* used the STPA-Sec to elicit systems security requirements for a notional autonomous space system. *Carter et al. (2018)* used STPA-Sec with a previous information elicitation process to analyze a small reconnaissance unmanned aerial vehicles. A further modeling technique has also been proposed by the same researchers to support a more efficient and traceable analysis (*Carter et al., 2019*). *Sidhu (2018)* applied an STPA-Sec extension with modified attack tree method to analyze cybersecurity of autonomous mining systems. *Wei & Madnick (2018)* analyzed an over-the-air software update use case in the automotive industry by using both STPA-Sec and CHASSIS and compared analysis outcomes.

The STPA-Sec is system-oriented and scenario-based. It has previously mentioned advantages of both system-oriented and scenario-based approaches. Comparing to another system-oriented scenario-based approach CHASSIS, the STPA-Sec views the system from the perspective of the control actions and addresses more strategic issues, while the CHASSIS analyzes the system from the functional use case aspects and pays more attention to tactical problems. Besides, the CHASSIS is more suitable for technical development phases since use cases and sequence diagrams are required when applying it. By contrast, STPA-Sec can be performed at a high system level in the concept phase, in which fewer technical documents are available.

Nevertheless, two limitations of the STPA-Sec have been found. First, it does not pay enough attention to the non-safety-related but security-critical issues, like data confidentiality. The first reason for causing such ignoring is that the security of the control action channels is not considered in the STPA-Sec. For example, in a water cooling system, "increase the water flow" is a typical action to control the system temperature. The STPA-Sec only analyzes insecure possibilities related to this action at a functional level but does not consider the possible insecure factors of the channels which deliver the
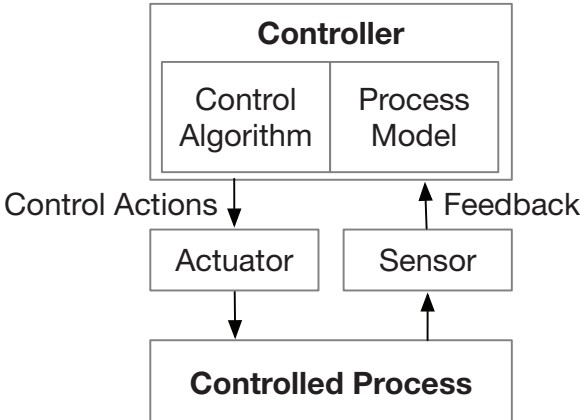

**Figure 1** **General control loop** (*Karatzas & Chassiakos, 2020*). General control loop structure of the STPA framework.                               

control information (or called commands) from the controller to the controlled process. Another reason for ignoring some security aspects is that some objects, which are not related to the control process (i.e., not presented in the control loop structure (Fig. 1)), are not considered. For example, in a vehicle software update system, "request" and "response" are the main control actions between the controller (e.g., an external tester) and the controlled process (e.g., electronic control units in a vehicle). The STPA-Sec mainly focuses on factors that may lead to losses related to these two control actions (e.g., an illegal request is accepted or a valid response is dropped). However, whether the data during the request or response is monitored illegally can not be identified directly by the STPA-Sec since the data confidentiality is not presented in the system STAMP model. The second limitation is that it lacks guidance when identifying information-security-related concepts including insecure behaviors and intended causal scenarios. The STPA-Sec inherits the STPA guide words to identify insecure control actions and uses components and interactions in the control loop as the "clues" to generate viable scenarios (*Young & Leveson, 2013*). Such safety-oriented identification methods may not efficient and direct to address information security problems.

Other researchers also pointed out similar limitations of the STPA-Sec. *Schmittner, Ma & Puschner (2016)* reported that the original STPA-Sec lacks guidance for intended causal scenarios, excludes considerations of the data exchange which is not directly connected to the process control and cannot cover more information-security centric properties such as confidentiality. *Torkildson, Li & Johnsen (2019)* also found that some essential security issues can be overlooked and recommended to strengthen STPA-Sec by combining it with data-flow-based threat models. However, Torkildson's approach converts the STPA control structure into a data flow diagram by simply replacing control action and feedback paths with data channels. Although such a data flow diagram helps to identify more data-related threats than using STPA-Sec alone, this diagram based on the original control loop does not view the system from the data aspect initially and may also overlook security issues that are not related to the control processes.

Furthermore, the STPA-Sec approach regards the security issue as one of the key threats affecting system safety (*Wei & Madnick, 2018*) and only supports the identification of safety-related security goals (*Martin et al., 2017*). Non-safety-related security issues like confidentiality may be overlooked.

To sum up, the STPA-Sec can address safety-related security problems, while the proposed STPA-DFSec reorients the scope of the STPA-based security analysis approach to consider more non-safety-related but information security problems. Furthermore, efficient guidance is needed to better support such information security analysis based on the STPA framework.

## METHODOLOGY

### Brief introduction of STPA and STPA-Sec

We briefly introduce the original STPA framework as the basis of the proposed approach in this section.

STPA starts with defining the purpose of the analysis, including system-level losses, hazards, and constraints. Losses are about something valuable and unacceptable to the stakeholders. A hazard is a system state or set of conditions that, together with a particular set of worst-case environmental conditions, will lead to a loss. Finally, system constraints can be derived from identified hazards, which specify system conditions or behaviors that need to be satisfied to prevent hazards and ultimately prevent losses (*Leveson & Thomas, 2018*).

Then, the control structure needs to be built to describe relationships and interactions by modeling the system as a set of control loops (show in Fig. 1).

The third step is to identify unsafe control actions, which will lead to a hazard in a particular context and worst-case environment (*Leveson & Thomas, 2018*), based on the previously built structure. Four ways of being unsafe are provided in STPA as guide words for the identification.

Finally, loss scenarios, which describe the causal factors that can lead to unsafe control actions, are identified. Two types of loss scenarios must be considered, which are "scenarios that lead to unsafe control actions" and "scenarios in which control actions are improperly executed or not executed" (*Leveson & Thomas, 2018*). Each identified scenario represents a system failure that needs to be controlled by designers.

The STPA-Sec, as an extension for security considerations, shares the same basic steps. Vulnerabilities, instead of hazards, are identified in the first step since vulnerabilities lead to security incidents, which is just like hazards lead to safety incidents (*Young & Leveson, 2013*). Similarly, final identified loss scenarios represent system vulnerabilities that need to be controlled.

### STPA-DFSec approach

The proposed STPA-DFSec follows the general STPA framework but introduces a data-flow-based structure for information security considerations. The main steps are described as follows.

**Table 2 General list of losses.** Recommended high-level losses for the initial loss identification step.

| Label | Loss | Description |
|---|---|---|
| L-1 | Loss of life or cause injury to life | Includes human and animal life |
| L-2 | Loss of physical property | Represents physical objects belonging to stakeholders (e.g., devices) |
| L-3 | Loss of non-physical property | Represents virtual properties belonging to stakeholders (e.g., sensitive information, reputation) |
| L-4 | Loss of environment | Includes the natural or artificial world |

### Step 1: Define the purpose of the analysis

Being similar to the STPA-Sec, the first step is to identify system-level losses, vulnerabilities, and constraints to figure out what are unacceptable results that need to be avoided at the system strategy level.

To help to identify losses, the STPA-DFSec provides general guidance for loss identification based on the study (*Yu & Luo, 2020*) of various safety- and security-related definitions from standards and technical documents in industries like ISO 26262 (*ISO, 2018*) and SAE J3061 guidebook (*SAE, 2016*). All possibilities of losses at a high abstract level are listed in Table 2. The loss list of a particular case is a subset of this general list and can be described concretely according to practical situations.

Vulnerabilities are system weaknesses that may lead to losses under external force. General security attributes like confidentiality, integrity, and availability (called C.I.A. Triad Model) can be used as guide words to classify the security problems and elicit potential vulnerabilities.

Finally, system-level constraints can be easily obtained by simply inversing the vulnerabilities or describing how to minimize the losses if the vulnerabilities are exploited (*Leveson & Thomas, 2018*).

### Step 2: Model functional interaction structure

Instead of the control structure, a Functional Interaction Structure (FIS) based on data flows is created to interpret how the system works from the perspective of data flows. We choose the data-flow-based diagram because data is the carrier of information. To view a system from the perspective of data flows is a more direct and efficient way to consider information security problems.

A component can be decomposed into a set of functions. Each function is a basic data process unit to handle the input data and output processed data. Data flow channels are the bridges between different functions to exchange information to finally accomplish overall system missions. The interactions between FIS components are viewed as the data exchanges between peer functions in different components. The data flow channels between different components are via the physical communication channels (e.g., cables and wireless channels), while the interactions among functions in one component go through the logical channels (e.g., via global variants and function parameters).

A function works based on its inputs and algorithms and outputs results. The processing environment, together with inputs and algorithms, will affect function behaviors and final outputs. Inputs and outputs, instead of control actions and feedback,

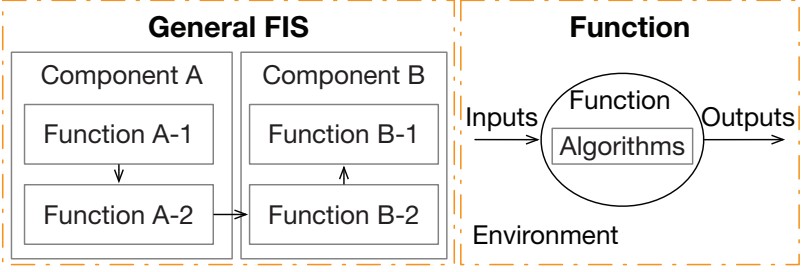

**Figure 2 General FIS and its basic element "Function".** General functional interaction structure of the proposed approach and the details of its basic element (called "function").

**Table 3 Enumerated common functions for FIS.** Enumeration of commonly used functions in the functional interaction structure, including data process and communication functions and cryptography-related functions.

| Classes | Functions |
|---|---|
| Data process and communication functions | process plain data; transmit data to; receive data from; validate received data (according to specifications, e.g., format and value range) |
| Cryptography-related functions | encrypt plain text by x; decrypt cipher text by x; calculate signature; validate signature; calculate MAC, validate MAC; other functions (e.g., key management related function, depends on available information) |

are interactions between components in FIS. Figure 2 shows a general interaction structure with arrows as data flows and the elements of a function.

The FIS is created based on high-level system description files, which describe the overall system purpose, general architecture with main components, and functional requirements related to the data process. How concrete an FIS is depends on how detailed the description files are.

Functions in an FIS are identified from the perspective of the data process. A common function set (see Table 3) is provided to help the establishment of the system FIS. The enumerated cryptography-related functions are derived from the cryptographer's toolbox, which consists of six kinds of cryptography mechanisms (symmetric key algorithm, asymmetric key algorithm, message authentication code, digital signature, one-way hash function, and random number generation) (*Schneier, 2003*). Users can pick the enumerated functions to build their system FIS at a general level or refine the function in particular cases if detailed information is available. Other possible case-specific functions derived from the system descriptions can also be added to the structure. This function database can be extended and refined by the development team to make the database more practical for particular design domains. For example, the "transmit data to" function can be refined as "transmit data via CAN bus to" or "transmit data via WiFi to" if the communication channel between components is known at this design stage. Different communication media has different initial security vulnerabilities which can lead to further specific analysis. The "encrypt plain text by x" can be refined as "encrypt plain text by AES-128-CBC algorithm" or "encrypt plain text by AES-256-CBC

**Table 4 Template for identifying IFBs.** Template for identifying insecure function behaviors with three guide words.

| Function (F) | GW: Not being Executed Causes Vulnerabilities (NECV)[1] | GW: being Executed Causes Vulnerabilities (ECV)[2] | GW: being executed but Exceeding Time Limits causes vulnerabilities (ETL)[3] |
|---|---|---|---|
| $S\_F_n$ | $S\_F_n\_IFB\__m$[4] | $S\_F_n\_IFB_{m+1}$ | $S\_F_n\_IFB_{m+2}$ |
| (e.g.) "Encrypt data" function | Function is bypassed but returns a fake OK result. | Data is encrypted by a forged key (provided by attacker). | Data encryption violated the process time limit. |

**Notes:**
[1] Adapted from the STPA UCA guide word "not providing causes hazard".
[2] Adapted from the STPA UCA guide word "providing causes hazard".
[3] Adapted from the STPA UCA guide words "too early, too late, out of order" and "stopped too soon, applied too long".
[4] $S\_F_n\_IFB_m$ is the IFB label, in which S represents the subject of the function.

algorithm". The execution time of these functions are different and should be considered when identifying timing-related losses. Therefore, by the refinement, analysts can further consider the security vulnerabilities related to the concrete function characteristics (e.g., transmission media or cryptographic algorithms) and obtain more specific outcomes finally.

Note that when identifying functions and their interactions, analysts only need to focus on the system function aspect. Potential attributes of functions or interactions like the real-time capacity do not need to be considered in this step. Possible threats to the system security caused by such attributes will be addressed in later steps. For example, we do not consider the real-time performance when constructing an FIS. However, the insecure functional behavior related to the poor real-time capacity of a function or interaction will be identified by using the guide word "Being executed but exceeding the time limits causing vulnerabilities".

### Step 3: Identify insecure function behaviors

From the established FIS, Insecure Function Behaviors (IFB), which are behaviors that will lead to a system vulnerability in a particular context like a worst-case environment, can be identified with the help of guide words. Table 4 is the template for identifying IFBs with guide words (GW) adapted from the STPA Unsafe Control Action (UCA) ones.

For identifying the "not being executed causes vulnerabilities" IFBs, it is helpful to consider if a function has the possibility of being bypassed or rejected but pretending to have been executed correctly. For identifying the "being executed causes vulnerability" IFBs, the possible weaknesses of the function execution conditions (e.g., inputs and execution context) are considered. As for the "being executed but exceeding time limits causes vulnerabilities" IFBs, whether the timeout will lead to the vulnerabilities is taken into account. How detailed an IFB is described depends on the information available. In any case, the analysis can be done with basic system information at a high system level.

### Step 4: Identify loss scenarios

Finally, Loss Scenarios (LS), as possible causes of IFBs, are identified. The guide words for identifying LSs are proposed based on the basic elements of a function. Table 5 is the template for identifying LSs with two classes of guide words. The "Function itself" class helps to identify loss scenarios with causes from the function side, while the "Execution environment (Env)" class refers to loss scenarios caused by the external factors.

**Table 5 Template for identifying LSs.** Template for identifying loss scenarios with five guide words.

| IFBs | GW: Function Itself | GW: Env- Function Inputs | GW: Env- Calling Behaviors | GW: Env- Computing Resources | GW: Env- Links |
|---|---|---|---|---|---|
| $S\_F_{n\_}\ IFB_m$ | $S\_F_{n\_}\ IFB_m\_LS_p$ | $S\_F_{n\_}\ IFB_m\_LS_{p+1}$ | $S\_F_{n\_}\ IFB_m\_LS_{p+2}$ | $S\_F_{n\_}\ IFB_m\_LS_{p+3}$ | $S\_F_{n\_}\ IFB_m\_LS_{p+4}$ |
| (e.g.) Encryption process violates the time limits | Process algorithm is modified, which leads to the timeout | Input size exceeds the limits but is not detected | / | Computing resource is occupied by others maliciously | / |
| (e.g.) Data leaks in the transmission | No or inadequate anti-leakage mechanism is used | / | / | / | Transmission channel is monitored illegally |

**Table 6 Summary of STPA-DFSec and STPA-Sec steps.** Summary and comparison of STPA-DFSec and STPA-Sec approaches with differences marked.

| Basic four steps | STPA-DFSec details | STPA-Sec details |
|---|---|---|
| Step 1: Define the purpose of the analysis | Identify system-level losses, vulnerabilities, and constraints. Link vulnerabilities with corresponding losses and security attributes[+]. A general losses list is provided[+] | Identify system-level losses, vulnerabilities, and constraints |
| Step 2: Model the system structure | Model the system by functional interaction structure based on data flows[*]. A common function set for FIS is provided[+] | Model the system by functional control structure based on the control loop |
| Step 3: Identify insecure items | Use adapted guide words[*] ("not being executed", "being executed" and "being executed but exceeding the time limits") to identify insecure function behaviors | Use guide words ("not providing", "providing", "too early, too late, out of order", "stopped too soon, applied too long") to identify insecure control actions |
| Step 4: Identify loss scenarios | Use adapted guide words[*] ("function itself", "execution environment (incl. function inputs, calling behaviors, computing resources, and links)") to identify loss scenarios | Use guide words ("unsafe controller behavior", "inadequate feedback and information", "involving the control path", "related to the controlled process") to identify loss scenarios |

**Notes:**
[+] Added features of the STPA-DFSec.
[*] Modified steps in comparison with the original STPA-Sec.

Each loss scenario represents a system vulnerability that should be controlled by designers or operators. Detailed system constraints can also be derived from loss scenarios by inversing the conditions of loss scenarios or defining what the system must do in case the incident occurs. System constraints are inputs for further design phases.

### Summary

Table 6 summarizes the process steps of both the STPA-DFSec and STPA-Sec with highlights of differences, in which "+" denotes added features of the STPA-DFSec and "*" represents modified steps in comparison with the original STPA-Sec.

## CASE STUDY

### Use case definition and assumption

In this section, a Bluetooth digital key system of a vehicle is introduced as a toy example, which consists of three main physical components and aims to lock or unlock vehicle doors by a smartphone. Communication between different entities via wireless channels are protected by cryptographic mechanisms. The high-level sequence diagram of two main services, as the input of the analysis, is shown in Fig. 3 to describe how the

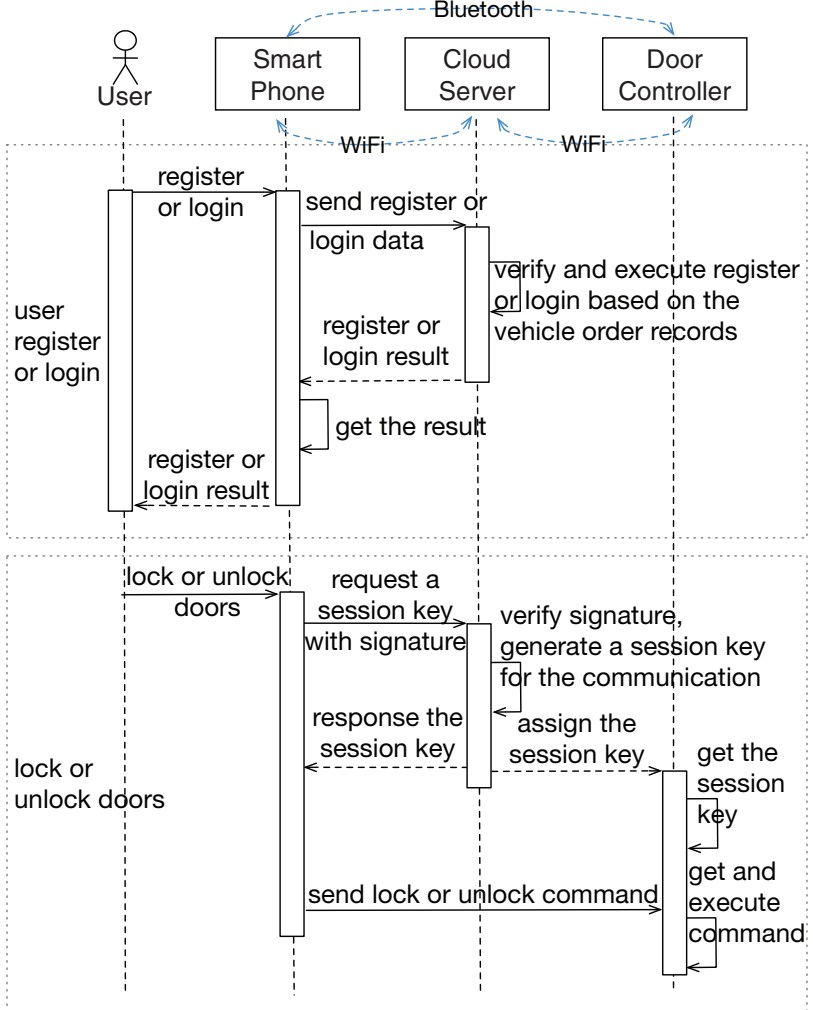

**Figure 3 Sequence diagram of the example system.** Sequence diagram of two services of the example system, which are "user register or login" and "lock or unlock doors" services.

system works. The asymmetric key algorithm is used in the communication with the cloud server, and the symmetric algorithm is used in the data transmission between the smartphone and the door controller. The research question in this case is: what are the security requirements of such an information-security-critical system?

In this example, we assume that the connections between components have been established, but the connection is not ensured to be secure. The public and secret keys required for the cryptographic algorithms have been prepared in advance. In this research, we only focus on security issues, which means that the system can work properly without intended external disturbances and the system development errors and hardware random failures are out of the scope.

## Analysis by STPA-DFSec
The STPA-DFSec analysis of the example system is presented in this section.

**Table 7 Losses, vulnerabilities, and constraints of the use case.** Identified system-level losses, vulnerabilities, and constraints of the example case, with trace information in the bracket.

L-1: Loss of physical property (incl. the vehicle and properties in it)

L-2: Loss of non-physical property (incl. manufacturer's reputation and intellectual property)

V-1: Doors can be controlled by invalid users, which is not detected by valid users (e.g., A theft opens the door without being noticed.) [L-1/2, Integrity]

V-2: Doors can not be controlled by valid users (e.g., Car owner can not lock the door when parking.) [L-2, Availability]

V-3: Sensitive information (e.g., communication protocol and personal data) is leaked. [L-2, Confidentiality]

SC-1: Doors should not be controlled by invalid users [V-1]

SC-2: If doors are controlled by invalid users, it must be detected and recovered [V-1]

SC-3: Doors should always be controlled by valid users [V-2]

SC-4: If doors can not be controlled by valid users, it should be fixed within an acceptable period [V-2]

SC-5: Sensitive information should be protected from leakage [V-3]

SC-6: If sensitive information is leaked, it should be detected and reactions need to be taken to minimize losses [V-3]

In step 1, the system-level losses (L) are firstly identified according to the provided general losses list (Table 2) as well as the purpose and functionalities defined in the system specification. Then, vulnerabilities at the system level are considered. In this case, we use the C.I.A. security attributes (i.e., confidentiality, integrity, and availability) as the identification guide words to identify vulnerabilities in these three aspects. The related security attributes, together with the linked losses, are listed in the blankets after the vulnerability descriptions. Finally, we convert the vulnerabilities into system constraints by directly inversing the vulnerable situations and describing what should be done if the vulnerability exploits. All the mentioned system-level losses, vulnerabilities, and constraints are listed in Table 7.

In step 2, the functional interaction structure is created based on the system data flows (shown in Fig. 4). We pick general functions in the proposed function database (Table 3), add some specific information related to this concrete system, and link them with data flow arrows based on the system sequence diagram (Fig. 3). In contrast to most traditional approaches, this analysis includes participants (user and manufacturer) outside the physical system boundary. Functions in a human or a manufacturer can also be refined into more detailed movements like "make a discussion", "press a button", or "manage the passwords". Since we mainly focus on the physical part in this analysis, human movements are simplified as one "human operation" function.

In step 3, insecure functional behaviors are identified by using the proposed guide words for each function. To increase the readability of this example case, the IFB labels of this demonstration are created in the form of "subjectName_Func%d_IFB%d" (%d represents a number). For example, the first identified IFB of a function is notated as "Phone_Func1_IFB1" (or "P_F1_IFB1" for short as presented in the template (Table 4)). Identified IFBs of an example function are shown in Table 8.

In step 4, we consider the potential causes of previously obtained IFBs with the help of guide words and possible previous experience to identify loss scenarios of each IFB.

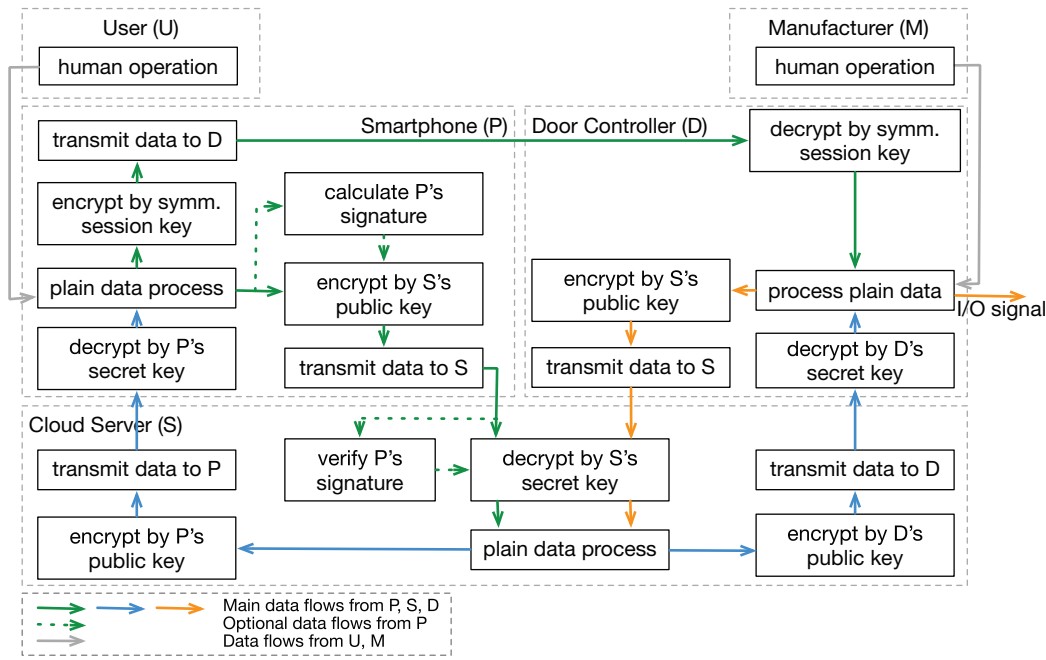

**Figure 4 Functional interaction structure based on data flows.** Functional interaction structure of the example case, including five system components and their data flow interactions.

**Table 8 Identified insecure function behavior examples.** Identified insecure function behaviors of the example function "encrypt data by S's public key".

| Function | GW: NECV | GW: ECV | GW: ETL |
|---|---|---|---|
| Phone_ Func1 | Phone_Func1_ IFB1 | Phone_Func1_IFB2, Phone_Func1_IFB3 | Phone_Func1_ IFB4 |

**IFB Description:**

Phone_Func1: "encrypt data by S's public key" function

Phone_Func1_IFB1: Data is not encrypted, but the function is pretended to have been executed correctly [V-1/3]

Phone_Func1_IFB2: Data is encrypted by a forged public key [V-1/3]

Phone_Func1_IFB3: Data is encrypted with malicious algorithms [V-1/3]

Phone_Func1_IFB4: Encryption process takes too long to violate the protocol time limits, which aborts the expected mission [V-2]

Similarly, LS notations are created in the form of "subjectName_Func%d_IFB%d_LS%d" (%d represents a number). For example, "Phone_Func1_IFB1_LS1" (or "P_F1_IFB1_LS1" for short as presented in the template (Table 5)) is the notation of the first loss scenario of the IFB labeled "Phone_Func1_IFB1". Examples of LSs related to IFBs in Table 8 are listed in Table 9.

## Analysis by STPA-Sec

We also analyzed the example system by the STPA-Sec. Due to the same system model, the system-level losses, vulnerabilities, and constraints are the same as those in the STPA-DFSec analysis. Therefore, the work here starts with drawing the system control

**Table 9 Loss scenarios of IFBs.** Identified loss scenarios of the listed IFBs in Table 8.

| IFBs | GW: function itself | GW: environment |
|---|---|---|
| Phone_Func1_IFB1 | / | Phone_Func1_IFB1_LS1 |
| Phone_Func1_IFB2 | / | Phone_Func1_IFB2_LS1 |
| Phone_Func1_IFB3 | Phone_Func1_IFB3_LS1 | / |
| Phone_Func1_IFB4 | Phone_Func1_IFB4_LS1 | Phone_Func1_IFB4_LS2, Phone_Func1_IFB4_LS3 |

**LS Description:**

Phone_Func1_IFB1_LS1: Function is bypassed but returns a fake OK result

Phone_Func1_IFB2_LS1: Valid key is replaced by a forged one

Phone_Func1_IFB3_LS1: Algorithm is maliciously modified by the attacker

Phone_Func1_IFB4_LS1: Algorithm is maliciously modified by the attacker, which requires more computing resource

Phone_Func1_IFB4_LS2: Input length exceeds the limitation but is not detected

Phone_Func1_IFB4_LS3: Computing resource is occupied by others maliciously

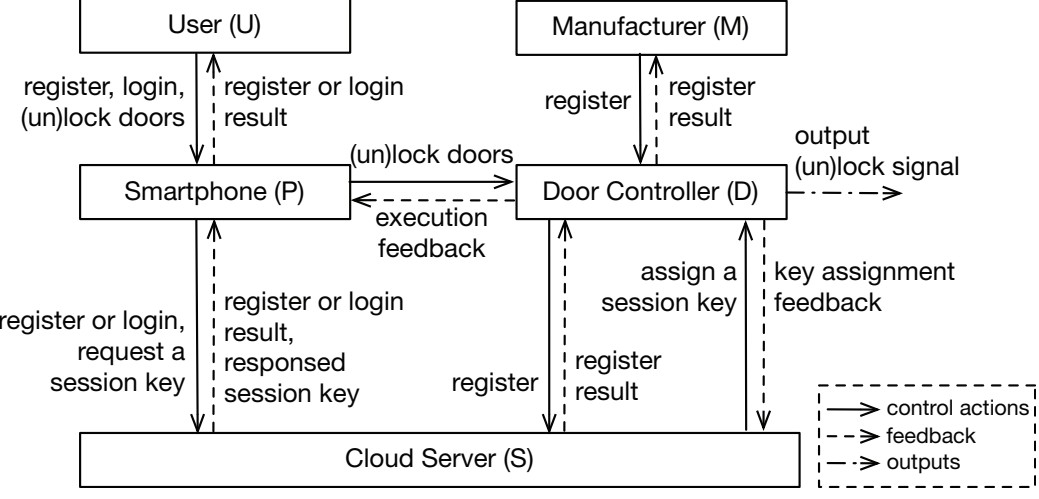

**Figure 5 Control structure of the system.** Control structure of the example case, including five system components and their control loop interactions.

**Table 10 Identified insecure control actions.** Identified insecure control actions of the example action "smartphone registers at the server".

| Control action | GW: not providing causes vulnerability | GW: providing causes vulnerability | GW: timing issues cause vulnerability[1] |
|---|---|---|---|
| Phone_CtrlAction1 | Phone_CtrlAction1_ Insec1 | Phone_CtrlAction1_ Insec2 | / |

**Label Description:**

Phone_CtrlAction1: Smartphone registers at the server (i.e., send account ID, password and smartphone's public key to the server)

Phone_CtrlAction1_Insec1: Smartphone does not register at the server correctly [V-2]

Phone_CtrlAction1_Insec2: Register is done successfully, but sensitive information (account ID and password) is leaked [V-3]

**Note:**

[1] The guide word "timing issues cause vulnerability" represents "too early, too late, out of order" and "stopped too soon, applied too long" in the STPA-Sec.

**Table 11 Loss Scenarios of ICAs.** Identified loss scenarios of the listed ICAs in Table 10.

| Insecure control action | GW: controller | GW: controller path | GW: controlled process | GW: feedback path |
|---|---|---|---|---|
| Phone_CtrlAction1 _Insec1 | Phone_CtrlAction1 _Insec1_LS1 | Phone_CtrlAction1 _Insec1_LS2 | Phone_CtrlAction1 _Insec1_LS3 | Phone_CtrlAction1 _Insec1_LS4 |
| Phone_CtrlAction1 _Insec2 | Phone_CtrlAction1 _Insec2_LS1 | Phone_CtrlAction1 _Insec2_LS2 | Phone_CtrlAction1 _Insec2_LS3 | / |

**LS Description:**

Phone_CtrlAction1_Insec1_LS1: Smartphone's software is modified maliciously

Phone_CtrlAction1_Insec1_LS2: The control command is blocked on the path

Phone_CtrlAction1_Insec1_LS3: Server's software is modified maliciously

Phone_CtrlAction1_Insec1_LS4: Register is done correctly but returns a NOK result

Phone_CtrlAction1_Insec2_LS1: No data protection mechanism is used at the smartphone

Phone_CtrlAction1_Insec2_LS2: Data is eavesdropped and decrypted at the path

Phone_CtrlAction1_Insec2_LS3: No data protection mechanism is used at the server

structure shown in Fig. 5, and then Insecure Control Actions (ICA) are identified. Examples of ICAs are shown in Table 10, in which the first letter of the label represents the control action's controller.

Finally, loss scenarios of each ICAs are identified. Examples of LSs are shown in Table 11.

## Outcome comparison

Functions and control actions are basic elements in the STPA-DFSec and STPA-Sec respectively. Normally, a control action includes several functions to provide a service. For example, the control action Phone_CtrlAction1 (Smartphone registers at the server.) consists of several functions including "plain data process", "transmit data to S" and "encrypt data by S's public key" and so on. Therefore, the relationship between these two elements is that a sequence of the execution of functions forms a control action.

To find out how both approaches work on the same use case, we mapped the analysis outcomes of both analyses. We found that a loss scenario identified by the STPA-Sec can be mapped to several STPA-DFSec loss scenarios. For example, the loss scenario Phone_CtrlAction1_Insec1_LS1 (Smartphone's software is modified maliciously) can be mapped to Phone_Func1_IFB1_LS1 (Function is bypassed but returns a fake OK result.), Phone_Func1_IFB2_LS1 (Valid key is replaced by a forged one.), and Phone_Func1_IFB3_LS1 (Algorithm is maliciously modified by the attacker.), which are all possibilities of causing losses related to the smartphone's software.

In reverse, an STPA-DFSec loss scenario can be related to several STPA-Sec ones because different control actions between components always share the same transmission channels and data process units. No matter what the control action is "register", "login" or "lock the door", they all require the "process plain data", "encrypt data by the key" and "transmit data" functions to perform the action.

In conclusion, the STPA-Sec focuses more on the application aspect of the system and aims to ensure the control actions secure, while the STPA-DFSec views the system as a

data process machine. It focuses on the security of the data process and flows and does not care the application meaning of the data.

## DISCUSSION

Finally, we discuss and conclude the differences and highlights of both approaches. The STPA-DFSec focuses on information flows and discusses possible vulnerabilities along the path of data flows, which helps to identify more detailed loss scenarios from the perspective of information flows. By contrast, the STPA-Sec can reveal more insecure details linked to concrete application scenarios since control actions are derived from the application aspect. The STPA-DFSec addresses in which data process unit a loss scenario occurs, while the STPA-Sec addresses in which application scenario a loss scenario occurs.

To choose which approach to use depends on particular cases. Two principles can be used to help the decision. The first one is according to the system purpose. If the analyst focuses on the data process and transmission security, the STPA-DFSec is more suitable for the analysis from the data side. If providing proper and secure services is the main object, the STPA-Sec is applicable to identify insecure issues linking with application scenarios. The second principle is to consider who uses it. The STPA-DFSec is suitable for designers who are responsible for technical structure and design, while STPA-Sec is more useful for ones who design the system functionalities and make more high-level decisions.

Note that the proposed approach does not rely on the known network vulnerabilities or attacks (e.g., eavesdropping and spoofing) since it is system-oriented and not threat-oriented. However, the known vulnerabilities can be used as clues when identifying system-level vulnerabilities in step 1. For example, "V-1" in Table 7 is actually a spoofing attack but does not use the word "spoofing" explicitly, and it is identified by the guide word "integrity". The known vulnerabilities can be kept in mind in the whole analysis process and work as the secondary clues for identifying insecure behaviors or scenarios. However, the experience from previous projects is not necessary for our approach. For example, the loss scenario "Transmission channel is monitored illegally" in Table 5, which may be attributed to an eavesdropping attack or other Denial of Service (DoS) attacks, is identified by the proposed steps and guide words and not by the known eavesdropping or DoS attacks. Analysts can use their preferred ways to describe the system vulnerabilities or choose proper identification guide words in the analysis. The known system vulnerabilities or attack types can help the identification but are not necessary conditions.

Two limitations of the STPA-DFSec have been identified. First, the STPA-based approaches lack the evaluation of identified scenarios. In practice, the resource for migrating insecure causes is limited. By evaluating and ranking the identified loss scenarios, the system designer can decide which insecure scenario should be avoided with high priority. To overcome this limitation, the STPA-based approaches can be used combining with other evaluation metrics (e.g., EVITA assessment method (*Ruddle et al., 2009*) and Common Vulnerability Scoring System (*FIRST, 2015*)) to assess the identified

insecure behaviors and scenarios. Second, the analyst should have corresponding information about the data processing of the target system (e.g., how data flows among components and what kind of data process function is contained). Otherwise, the functional interaction structure can not be constructed.

In practice, system security engineering is not able to ensure absolute security but provides a sufficient base of evidence that supports claims that the expected level of trustworthiness has been achieved (*Ross, McEvilley & Oren, 2016*). The analysis in security engineering is also not able to be proven complete. The completeness of the analysis and how detailed the results are normally depend on the analyst's knowledge and experience, design emphasis, and available system information. However, a proper systematic approach can help the analyst to be more confident in the analysis completeness (*Young & Leveson, 2014*). Proper guide words help to reduce the dependency on personal experience and subjective thinking and lead to relatively objective and valid results with less effort. The case study in this article represents the authors' understanding of the example system and works as a demo to show how to use the proposed approach.

## CONCLUSION

In this article, we have proposed a data-flow-based approach for security analysis of information-critical systems based on the STPA framework to overcome STPA-Sec's limitations. The analyses of a vehicle digital key system by using both the STPA-DFSec and STPA-Sec have been presented as examples to show how to use the approaches. Finally, we compared the analysis results, presented the differences between both approaches and discussed highlights and drawbacks of the proposed STPA-DFSec.

We have found that the proposed STPA-DFSec focuses on data flows and can reveal more details in information security aspects, which can not be addressed directly in the STPA-Sec analysis, while the STPA-Sec analyzes the system from the perspective of applications and concerns more safety-related security issues. Additionally, as an adaption of the STPA-Sec, the proposed STPA-DFSec, together with other STPA-based approaches, can be used to co-design complex systems in multi-disciplines under the unified STPA framework. Social aspects and human factors can also be included in the analysis, which are excluded from traditional analysis approaches.

The proposed approach is not a substitution but a complement to the original approach. By using STPA-DFSec, we view the system from a new perspective (i.e., the data processing aspect) other than control loops and may find new points that are not directly identified by the STPA-Sec. Because of the relationship between the control action and the function in both approaches, the identified insecure items and loss scenarios can be mapped to each other, which helps to understand and design the system better.

With the increasing connectivity and complexity of modern systems, more traditional safety-critical systems require information security to protect intellectual property or privacy nowadays (e.g., vehicles and healthcare devices). Based on the already-established system STAMP model of these systems, the proposed approach can be better integrated into the existing work and deal with the security aspect. Furthermore, compared to the existing approaches, the STPA-based ones are the better solutions that analyze the system

at a high strategy level, which provides a new point of the view for the security analysts to get possible new ideas or understanding of the target system.

In the future, we will study more real-world cases and conduct experiments with different groups of analysts to evaluate and refine the proposed approach in practice. Furthermore, we will formalize the analysis process and design tools to achieve analysis results automatically for higher working efficiency.

### Funding
This work was supported by the China Scholarship Council and funds of the German Federal Ministry of Education and Research under grant number 16KIS0995. The funders had no role in study design, data collection and analysis, decision to publish, or preparation of the manuscript.

### Grant Disclosures
The following grant information was disclosed by the authors:
German Federal Ministry of Education and Research: 16KIS0995.

### Competing Interests
Stefan Wagner is an Academic Editor for PeerJ.

### Author Contributions
- Jinghua Yu conceived and designed the experiments, performed the experiments, analyzed the data, prepared figures and/or tables, authored or reviewed drafts of the paper, and approved the final draft.
- Stefan Wagner conceived and designed the experiments, analyzed the data, authored or reviewed drafts of the paper, and approved the final draft.
- Feng Luo conceived and designed the experiments, authored or reviewed drafts of the paper, and approved the final draft.

### Data Availability
Data are available as a Supplemental File.

### Supplemental Information
Supplemental information for this article can be found online at http://dx.doi.org/10.7717/peerj-cs.362#supplemental-information.

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
