# Peer review of "Data-flow-based adaption of the System-Theoretic Process Analysis for Security (STPA-Sec)"

_PeerJ Computer Science, doi:10.7717/peerj-cs.362_

## Round 0.1 · original submission · Major Revisions

The article needs to be revised according to the comments of the reviewers and be returned within the next 30 days.

Reviewer 1 ·

Basic reporting

The paper claims that there are limitations with STPA-Sec predominantly because it lacks IT-centric information. It is not clear to me why this is a limitations, I would suggest changing the wording to instead claim that the STPA-DFSec method is an augmentation that reorients the scope of STPA-Sec. However, the authors don't address similar literature that has unified these two views using STPA:

A Preliminary Design-Phase Security Methodology for Cyber–Physical Systems, Systems

Systems‐theoretic security requirements modeling for cyber‐physical systems, Systems Engineering

An Ontological Metamodel for Cyber-Physical System Safety, Security, and Resilience Coengineering arxiv

A Top Down Approach for Eliciting Systems Security Requirements for a Notional Autonomous Space System

(This has little bearing on my assessment of the work and authors should not feel forced to cite this work; cite it only if you find it important to cite in the context of your work)

I believe the rest of the paper is okay in terms of basic reporting.

Experimental design

The work is described in sufficient detail, but I would like to see more of an application of the different transformation steps as in how is does the model start, get manipulated and end. Also I found some of the table notation confusing (e.g., P_CA1_ICA1) but I can't particularly think of a way of fixing this that is universal. I suggest these are modified in the paper to something more descriptive potentially with a quasi mathematical form (notice NOT mathematical merely using supscripts and superscripts – one of the benefits of STAMP is that it does not pretend to be mathematical).

Validity of the findings

I would like to see publicly accessible model that does the things (even if it does them in text documents) and see all the transformations in an online git repository. This would increase the likelihood that the results are valid and can be replicated in another system. This is also generally a lacking resource from STAMP so I think it could increase the visibility of the paper. If this is a toy example it should be said explicitly in the paper and still be offered publicly online in totality (I understand that paper appendices might do the same job but STAMP is difficult to enclose in tables in letter paper).

Additional comments

I personally like the idea and I think the paper illustrates the application of this work. I think as a potential augmentation to STAMP when IT-centric security properties are needed this is good work. But when is such a case? When one would use STAMP but want to capture confidentiality, integrity and availability? The paper could do a better job motivating this augmentation.

Reviewer 2 ·

Basic reporting

'No comment'

Experimental design

However some further clarifications should be given by the authors on a Functional Interaction Structure (FIS) based on data flows is created to interpret how the system works from the perspective of data flows.
- Further justification is needed for the interaction between the general FIS components and the related "functions".
-This interaction is done in real time?
-A better description for the how the function database can be extended and refined by the development team to make the database more practical and realisitic case studies.
-How the network vulnerabilities (presence of evendroppers, spoofing attacks, RF jamming etc.) affects the data-flow-based structure for information security considerations.
-Social aspects and human factors can also be included in the proposed analysis.

Validity of the findings

'No comment'

Additional comments

In this paper, the authors proposed a data-flow-based adaption of the STPA-Sec (named STPA-DFSec) to overcome the mentioned limitations and elicit security constraints systematically. The authors analytically desctribe the methodoly for the STPA-DFSec method to overcome the security-critical issues (e.g. data confidentiality) and identify the information security concepts. The main novelty of this paper is that the proposed methodology overcomes the original STPA-Sec limitations that it cannot cover more information-security centric properties such as confidentiality. The conclusions of this work are well sated and linked to original research.

However some further clarifications should be given by the authors on a Functional Interaction Structure (FIS) based on data flows is created to interpret how the system works from the perspective of data flows.
- Further justification is needed for the interaction between the general FIS components and the related "functions".
-This interaction is done in real time?
-A better description for the how the function database can be extended and refined by the development team to make the database more practical and realisitic case studies.
-How the network vulnerabilities (presence of evendroppers, spoofing attacks, RF jamming etc.) affects the data-flow-based structure for information security considerations.

---

## Round 0.2 · Minor Revisions

Please go through the article and perform some minor corrections in English as suggested by the reviewers. Also please refine figures 4 and 5 as suggested.

Reviewer 1 ·

Basic reporting

The paper should go through a (very) minor editorial review, quotation marks are the wrong way and some again very minor problems with English.

Figure 4 and 5 should be made bigger.

Everything else is acceptable for a paper on a high level systems theory paper like STPA.

Experimental design

Researchers show how they augment STPA and clearly state that it is a toy example so I am not expecting something more extensive.

Validity of the findings

The findings seem to follow the contributions stated in the paper.

Additional comments

No comment.

Reviewer 2 ·

Basic reporting

No comment

Experimental design

No comment

Validity of the findings

No comment

Additional comments

This paper proposes a data-flow-based adaption of the STPA-Sec (named STPA-DFSec) to overcome the mentioned STPA-Sec’s limitations. Μost of the reviewers' comments were covered by the authors' answers. The main motivation of this article is that the proposed STPA-DFSec follows the general STPA framework but introduces a data-flow-based structure for information security considerations. An additional contribution is that in the "Step 4 - Identification of loss scenarios" in which the possible causes of IFB are identified, new guidewords are defined for loss scenarios caused by external factors. However, the loss scenario ‘Transmission channel is monitored illegally’ in Table 5 can not be attributed to just eavesdropping attack but also to other Denial of Service (DoS) attacks.

---

## Round 0.3 · accepted · Accept

The article is publishable in its current form. I would suggest changing the past tense to present tense especially in the abstract.